# Dengue and Acute Pancreatitis: A Systematic Review

**DOI:** 10.3390/tropicalmed10120330

**Published:** 2025-11-24

**Authors:** Matteo Riccò, Antonio Cascio, Luca Pipitò, Marco Bottazzoli, Paolo Manzoni, Lilian Romina Brandolisio, Cecilia Nobili, Pasquale Gianluca Giuri

**Affiliations:** 1AUSL–IRCCS di Reggio Emilia, Servizio di Prevenzione e Sicurezza Negli Ambienti di Lavoro (SPSAL), Local Health Unit of Reggio Emilia, 42122 Reggio Emilia, Italy; 2Infectious and Tropical Diseases Unit, Department of Health Promotion, Mother and Child Care, Internal Medicine and Medical Specialties, “G D’Alessandro”, University of Palermo, AOUP P. Giaccone, 90127 Palermo, Italy; antonio.cascio03@unipa.it (A.C.);; 3Department of Otorhinolaryngology, APSS Trento, 38122 Trento, Italy; 4SCDU Pediatria e Neonatologia, Ospedale Degli Infermi, Ponderano (BI), Dipartimento Materno-Infantile, Local Health Unit of Biella (ASLBI), University of Torino, 10124 Torino, Italy; 5Department of Public Health and Pediatric Sciences, University of Torino School of Medicine, 10125 Turin, Italy; 6Department of Medicine and Diagnostics, AUSL di Parma, 43100 Parma, Italy

**Keywords:** dengue, acute pancreatitis, acute necrotizing pancreatitis, acute interstitial pancreatitis

## Abstract

Dengue fever typically presents as a febrile illness, and acute pancreatitis has been reported as a rare complication. Limited evidence exists regarding clinical features, imaging findings, and outcomes, particularly on the distinction between acute interstitial pancreatitis and acute necrotizing pancreatitis. This systematic review was therefore designed in accordance with PRISMA guidelines (PROSPERO ID: CRD420250631013) in order to characterize the clinical spectrum of dengue virus (DENV)-associated acute pancreatitis by recollecting available case reports from PubMed, EMBASE, Scopus, MedRxiv, and BioRxiv Case reports and case series, which were included if they described laboratory-confirmed DENV infection and met the diagnostic criteria for acute pancreatitis. Seventy cases of DENV-associated acute pancreatitis were identified, mostly from Asia (78.6%) and South America (17.1%). Patients were predominantly male (62.9%), with a mean age of 31.6 years. Acute interstitial pancreatitis was more common (65.7%) than acute necrotizing pancreatitis (34.3%). Acute necrotizing pancreatitis was associated with leukocytosis, pancreatic collections, multiorgan and respiratory failure, prolonged hospital stay, and higher mortality (25.0% vs. 4.3% for acute interstitial pancreatitis). Overall, the case fatality was 11.4%, and survival analysis demonstrated significantly worse outcomes for acute necrotizing pancreatitis. In summary, clinicians should maintain awareness of this complication, as early recognition and appropriate management may improve outcomes.

## 1. Introduction

### 1.1. Background: Dengue Virus

Dengue fever is caused by dengue virus (DENV), a small (50 nm diameter), positive-sense, single-stranded RNA (10.7 kilobases) virus belonging to the genus *Orthoflavivirus* in the *Flaviviridae* family [1,2]. The genome of DENV encodes for a single polyprotein that is then cleaved into three structural proteins (envelope, E; membrane, M; capsid, C) and seven non-structural proteins (NS1, NS2A, NS2B, NS3, NS4A, NS4B, and NS5) [1,3,4,5,6,7,8]. NS1 plays a key role in viral RNA replication, virion production, and immune evasion and represents a main diagnostic antigen [9]. DENV is usually classified into four serotypes sharing 65% to 70% of the amino acid sequence (DENV-1 to -4): DENV-1, -2, and -3, which are considered responsible for much of the global dengue burden, and DENV-4, which is usually associated with fewer outbreaks [2]. A fifth serotype (i.e., DENV-5) was detected in 2007, but its significance remains unclear [10,11]. Each serotype, in turn, includes several genotypes (i.e., a group of DENV isolates that have no more than 6% of nucleotide sequence divergence), and each genotype varies in terms of its distinctive viral fitness, virulence, epidemic potential [12,13], and even immune responses [14], with variable cross-lineage immunity.

### 1.2. Current Epidemiology of Dengue

Dengue is currently acknowledged as the most common mosquito-borne tropical viral disease in humans [1,2,5], as well as one of the most reported travel-associated infectious diseases [15]. Although there is substantial uncertainty due to the high proportion of asymptomatic infections and the inadequate reporting systems from several geographic areas [1,5,16,17], around 50% of all human beings are nowadays considered at risk for developing dengue infection due to the four main DENV serotypes (DENV 1-4) [10,12,13]. The evolutive success of DENV is due to a series of factors, including but not limited to the extensive and still increasing availability of competent mosquito vectors (i.e., *Aedes albopictus* and *Aedes aegypti*) [18,19]; the rapid, often uncontrolled urbanization of various geographic areas [10,13,20,21]; the increased importation of DENV from endemic areas to non-endemic areas by international travel [10,13,20,22]. In fact, the global incidence of dengue has steadily increased since the 1980s [10,13,14,20,21,23], swelling from less than 9 million cases in 1990 to 58.96 million cases in 2021 [24,25], and up to 100 million cases in 2024–2025 [2,24]. Nevertheless, the true number of annual infections may even exceed more recent estimates [2], possibly ranging from 284 to 528 million [14].

### 1.3. Clinical Features of Dengue

After the mosquito bite, DENV infects human mononuclear phagocyte lineage cells (i.e., monocytes, macrophages, and dendritic cells) [1,4] and then replicates by means of multiple molecular interactions that are ultimately involved in dengue pathogenesis [2]. Due to a wide range of potential cell receptors, DENV and dengue antigens have been identified in many tissues, including the liver, spleen, lymph node, thymus, lung, and skin [26], contributing to the quite heterogenous clinical features associated with DENV infection. While 75% of incident cases of DENV infection remain asymptomatic [27], following the incubation period (2 to 14 days), around 25% of patients develop an abrupt onset of symptoms, which in the past led to the dichotomization into dengue fever (an acute, self-limiting, and disabling illness with high fever (up to 40 °C), nausea, vomiting, rash, and joint pain) [10,20,21] and dengue hemorrhagic fever/dengue shock syndrome (DHF/DSS), a severe, life-threatening condition [1,5] (Appendix B, Figure A1) characterized by the diffuse and extensive damage of lymph nodes and blood arteries [1]. The ensuing capillary damage results in capillary leak syndrome, subsequently leading to multiorgan failure and potentially shock, with or without associated hemorrhagic manifestations, with an eventual mortality peaking to 20% [14]. Complications of dengue can be elicited by all DENV serotypes and genotypes, following a complicated interplay between the virus and the immune system, which has been only partially explained [1,4,5,16,28].

Since 2009, symptomatic dengue cases have been reclassified by the World Health Organization (WHO) into dengue (i.e., a clinical syndrome among individuals living in or who have traveled to a dengue-endemic area), including any of the following signs/symptoms: nausea, vomiting, rash, aches and pains, a positive tourniquet test, leukopenia, or any symptom with or without warning signs (i.e., abdominal pain or tenderness, persistent vomiting, clinical fluid accumulation, mucosal bleeding, lethargy, restlessness, and liver enlargement) and severe dengue. Severe dengue has been otherwise defined as dengue with any of the following clinical manifestations: severe plasma leakage; shock; fluid accumulation; respiratory distress features; multiorgan failure (including diagnosis of transaminases ≥ 1000 IU/L); impaired consciousness; and/or heart impairment [4,29,30] (Appendix B, Figure A2).

### 1.4. Dengue and Pancreatitis: What We Currently Know

With an estimated annual incidence rate of 34.8 per 100,000 [31], acute pancreatitis is considered a common disease with significant morbidity and mortality [32]. Conversely, DENV-associated pancreatitis is considered a quite rare or even extremely rare condition [33,34]. On the one hand, the direct infection of pancreatic cells remains unproven. Immunofluorescence studies have previously revealed the presence of the DENV protein NS3 in macrophages associated with pancreatic tissue [35]. On the other hand, a few cases have been previously described in the international literature and are included in two systematic reviews on the infectious causes of acute pancreatitis [33,36]. Even the clinical features of DENV-associated pancreatitis remain doubtful, and this knowledge gap could contribute to the eventual underestimation of this condition, particularly during outbreaks or epidemic episodes. While in their earlier report, Estofolete et al. [37] suggested that DENV infection in acute pancreatitis is associated with severe clinical features from DHF/DSS and abnormal serologic indicators, such as hypocalcemia, hyperlipasemia, hyperglycemia, and hypoamylasemia, it remains unclear whether DENV-associated acute pancreatitis may be more frequently characterized as acute interstitial pancreatitis (AIP) or acute necrotizing pancreatitis (ANP). Acute interstitial pancreatitis is the diffuse, non-necrotizing inflammation of the pancreas that accounts for around 90% of all cases of acute pancreatitis [38,39]. In contrast-enhanced computed tomography (CECT) imaging, the gland usually displays either focal or diffuse enlargement, with a generally homogenous enhancement, and peripancreatic fat may appear stranded, with small quantities of peripancreatic fluids [39,40,41]. Acute interstitial pancreatitis is mostly a benign condition, with symptoms that tend to resolve in the first week. Acute necrotizing pancreatitis, by contrast, accounts for from 5% to 10% of all cases of acute pancreatitis and is associated with higher morbidity and mortality. In CECT, ANP is usually associated with intrapancreatic and/or peri-pancreatic hypodensities (i.e., areas of necrosis), which can encompass not only the whole of the organ, and/or extensive collections within the organ, but also other anatomical regions of the abdomen [38].

The present systematic review was therefore designed with the aim of assessing whether or not acute pancreatitis associated with DENV infection has any distinctive clinical feature. To summarize and reconcile the available data on acute pancreatitis during DENV infection, we specifically targeted all published case reports providing detailed clinical and radiological features, focusing on cases developing AIP compared to ANP.

## 2. Materials and Methods

### 2.1. Study Selection and Inclusion/Exclusion Criteria

We designed the present systematic review according to the PRISMA (Preferred Reporting Items for Systematic Reviews and Meta-Analyses) statement [42,43]. As a preliminary step, the research outline was recorded in the PROSPERO (Prospective Register of Systematic Reviews) database with the ID number CRD420250631013 (see PRISMA checklist in Appendix A).

The research concepts were defined according to the “PICO” strategy (Patient/Population/Problem; Investigated Results; Control/Comparator; Outcome) in terms of the following question: are the clinical features (I) and corresponding outcome (O) of documented cases of dengue virus infection (P) comparable in cases of interstitial and necrotizing pancreatitis (C)?

Three scientific databases (i.e., PubMed, Scopus, and EMBASE) and the preprint repositories MedRxiv and BioRxiv were searched for all entries on dengue virus infection and pancreatitis, without any chronological restriction. The research strategy included the research terms in Table 1, which were then refined in accordance with the specific requirements of the inquired search platforms.

Furthermore, a “snowball” approach (i.e., screening all the retrieved articles for suitable additional citations) was implemented to identify all case reports and case series not reported by the direct inquiry of databases; in other words, all references of the retrieved studies were accurately analyzed, and suitable entries were specifically searched to retrieve any potential reports.

For the aims of this review, original research publications were included in the analyses if written in any of the languages spoken by the authors, i.e., English, Italian, German, French, Spanish, Portuguese, and Chinese, and when the main text provided detailed clinical characteristics of individual cases of acute pancreatitis in subjects affected by a documented dengue virus infection (see further details on inclusion criteria).

To be considered consistent with this review and therefore included in the present systematic review, the following inclusion criteria had to be fulfilled:(1)Availability of detailed clinical and/or imaging features of the reported case.(2)Diagnosis of dengue virus infection via detection of IgG/IgM antibodies in a serum sample and/or detection of NS1 antigen and/or detection of viral RNA by means of polymerase chain reaction (PCR) assay and/or isolation of dengue virus from the index patient. Only for reports published before the widespread commercial availability of the enzyme-linked immunosorbent assay (ELISA) (i.e., mid-1980s), hemagglutination tests were used to document dengue virus infection; studies were deemed eligible for inclusion in the systematic review even when the specific diagnostic test was not explicitly reported, provided that the original source confirmed a laboratory-established diagnosis (i.e., “not provided”, NP).(3)Diagnosis of acute pancreatitis by means of two of the following features: (a) abdominal pain of acute onset, severe, affecting the epigastric region, and often radiating to the back; (b) serum lipase and/or amylase activity at least three times greater than the upper limit of the normal; (c) documented findings of acute pancreatis on contrast-enhanced computed tomography and/or magnetic resonance imaging and/or transabdominal ultrasonography [38,39,40,41].

Similarly, we deliberately excluded cases from the review where (1) laboratory diagnosis of DENV infection was not reported; (2) diagnosis of acute pancreatitis was only based on clinical features, without any laboratory or imaging features. Cases deprived of laboratory confirmation tests were excluded from the pooled analysis in order to focus the present systematic review on the clinical features of dengue-associated acute pancreatitis. While the positive predictive value of clinical criteria is high, particularly for illnesses meeting all criteria for DHF from within endemic areas, the potential burden associated with outbreaks, limited resources, false positives, and time of testing after symptom onset may impair the extensive application of laboratory testing and, in turn, the reliability of reported clinical features [2,44,45,46]. Moreover, DENV often co-circulates with other infections, such as yellow fever, chikungunya, and Zika (and, more recently, Oropouche virus in South America), which, in the early stages of infection, may mimic the features of dengue [44,45,47], and reports deprived of laboratory confirmation tests may therefore include cases of other non-DENV arbovirus infections.

The retrieved articles were then screened by their titles for their relevance to the subject [42,43]. Articles which were positively title-screened were eventually abstract-screened. If the content of the abstract was considered consistent with the aims and design of the present review, the corresponding full texts were independently assessed by two investigators (F.M., S.C.) and abstracted to obtain the following data:(a)Settings of the case: year, month or season, geographic region.(b)Age and gender of the reported cases.(c)Pre-existing clinical features, if any, and any of the following risk factors for acute pancreatitis [33,36,48,49,50,51,52,53]: alcohol abuse; smoking history; any previous diagnosis of gallstones; hypertriglyceridemia; endoscopic procedures; abdominal trauma; history of autoimmune disease; predisposing genetic mutations; any history of drugs associated with acute pancreatitis (angiotensin-converting enzyme (localized angioedema); statins (direct and accumulation toxicity); oral contraceptives or hormone replacement therapy, especially estrogen (hypertriglyceridemia, local arteriolar thrombosis); diuretics; antiviral therapy (HIV); valproic acid; and antidiabetic agents, such as GLP-1 mimetics).(d)Clinical characteristics: length of reported symptoms (days); fever (body temperature > 38 °C); abdominal pain; nausea; vomiting; any sign of altered consciousness; headache; skin rash; jaundice and scleral icterus; conjunctivitis; retro-orbital pain; pain; any sign of capillary refill disorder and peripheral edema; signs of minor bleedings (e.g., petechiae, purpura, subdermal hemorrhages, epistaxis, and gum bleeding); shortness of breath; hypotension (i.e., blood pressure < 90/60 mmHg); tachycardia (i.e., resting heart rate exceeding 100 beats per minute).(e)Diagnostic strategy for dengue virus infection (i.e., antigenic testing on viral NS1 protein, studies reporting IgM and IgG antibodies targeting DENV, polymerase chain reaction [PCR] studies).(f)Laboratory work-up results [54], specifically focusing on blood count values; blood sugar levels (BSLs); albuminemia; creatinine levels; liver function tests (i.e., serum aspartate transaminase [AST]; serum alanine aminotransferase [ALT]; alkaline phosphatase, gamma-glutamyl transferase [GGT]); lipases and amylases; values of proteins and/or lactate according to the normal range values of the parent institution.(g)Type of acute pancreatitis, i.e., AIP vs. ANP [38,39,40,41].(h)Features of all imaging studies, including ultrasonographic studies and CT scans.(i)Calculation of the modified CT severity index (CTSI), through a 0 to 10 score. When the CTSI was not directly available, it was calculated by means of the CT scans included in the source report [38,39,40,41,52,53,55,56].(j)Outcomes: hospital stay (days); whether any respiratory support was required or not; whether any surgery was performed, including drainage of collection and/or necrotic areas; survival vs. death; and weeks of total survival time.(k)If a certain patient was cross-posted by different studies, reports were accurately analyzed to fill the knowledge gaps and provide an extensive description of the clinical case, as well as to eliminate duplicates.

### 2.2. Qualitative Assessment

The qualitative assessment of the retrieved studies was performed following the recommendations from Murad et al. [57] for case reports and case series. According to these recommendations, each study was assessed through four domains (selection, ascertainment, causality of case, and reporting quality) by means of a series of binary items (“high risk” vs. “low risk”, for a corresponding score of 0 vs. +1): D1: “Does the patient(s) represent(s) the whole experience of the investigator (center) or is the selection method unclear to the extent that other patients with similar presentation may not have been reported?”; D2: “Was the exposure adequately ascertained?”; D3: “Was the outcome adequately ascertained?”; D4: “Were other alternative causes that may explain the observation ruled out?”; D5, “Was there a challenge/rechallenge phenomenon?”; D6, “Was there a dose–response effect?”; D7: “Was follow-up long enough for outcomes to occur?”; D8: “Is the case(s) described with sufficient details to allow other investigators to replicate the research or to allow practitioners to make inferences related to their practice?”. As D5 and D6 were specifically designed for studies on drug-related events, retrieved studies were rated zero to six through only six items (i.e., D1, D2, D3, D4, D7, and D8) by two investigators who independently read the full-text versions of eligible articles. Disagreements were tentatively resolved by consensus between the two reviewers. When the consensus was not obtained, input from a third investigator (M.R.) was sought and obtained. In order to recollect a large base of evidence, and in accordance with the original recommendations from Murad et al. [57], even studies with high-risk ratings in one or more of the assessed domains were included in the summary analysis.

### 2.3. Data Analysis

Included studies were primarily summarized by descriptive analysis. Where available, continuous variables were initially reported as means ± standard deviations. Continuous variables were initially tested for normality by means of the K2 test. Given the exploratory nature of the analysis and the presumptively small sample size, a significance level of α = 0.10 was used to increase the sensitivity to departures from normality. Therefore, a K2 test *p*-value of <0.100 did identify non-normally distributed variables, which were analyzed using Mann–Whitney or Kruskal–Wallis tests, while their correlation was analyzed by means of Spearman’s rank test. Variables with K2 test *p*-values of ≥0.100 were considered normally distributed, and their correlation was assessed by means of Pearson’s correlation coefficient. Levene’s test was then used to assess the equality of variances (i.e., homoscedasticity) of the data. Data exhibiting homoscedasticity and normality were compared between the groups by means of an independent-sample *t*-test. Heteroscedastic normal data were compared between the groups with the Welch test (unequal-variance *t*-test).

Categorical variables were reported as percent values, and their distribution was assessed through the chi-squared test concerning the dichotomous outcome variables of (1) diagnosis of acute necrotizing vs. acute interstitial pancreatitis; (2) reporting death during the follow-up. Due to the reduced sample size (less than 256 observations), Yates’s correction was applied to improve the accuracy of the *p*-value obtained.

Missing values for assessed variables were handled by means of listwise deletion; i.e., observations with incomplete data were excluded from the specific analysis in which the missingness occurred.

Each outcome variable was eventually included in two distinctive Cox regression models, which were designed as follows:-Model I, i.e., diagnosis of ANP vs. AIP, included the diagnosis of ANP as the outcome variable, and the time variable was represented by the age of the patient at the time of the onset of symptoms;-Model II, i.e., death of the patient during the follow-up, included the death of the patient as the outcome variable, and the time variable was represented by the time elapsed since the onset of the symptoms and the documented overall survival of the patient;-All other categorical variables that, in univariable analysis, were associated with the corresponding outcome variable with a *p*-value < 0.05 were also included in the stepwise binary logistic regression analysis model as explanatory variables.

Survival analyses (i.e., overall survival and disease-free survival) were eventually assessed through Kaplan–Meier statistics, including overall survival/disease-free survival by eventual status (death/relapse).

All statistical analyses were performed using IBM SPSS Statistics 26.0 for Macintosh (IBM Corp., Armonk, NY, USA), R (version 4.4.1), GraphPad Prism 10.5 (GraphPad Software LLC, Boston, MA, USA) and Rstudio (version 2025.05.0 Build 496; Posit Software, PBC; Boston, MA, USA) software, by means of the packages cutpointr (version 1.2.0) and fmsb (version 0.7.5).

## 3. Results

### 3.1. Results of the Query

The selection procedure is summarized in Figure 1.

Overall, a total of 66 articles and 70 case reports were considered consistent with the research questions and corresponding inclusion criteria and were therefore summarized and included in the analyses. Their characteristics are summarized in Table 2 (full details are provided in Appendix A). Briefly, only 2 case reports (2.9%) documented cases occurring before 2000 [58,59]; 8 between 2000 and 2009 (11.4%) [34,35,60,61,62,63,64]; 26 (37.1%) for the decade 2010–2019 [65,66,67,68,69,70,71,72,73,74,75,76,77,78,79,80,81,82,83,84,85,86,87,88,89,90]; while the remaining 34 cases were documented for the time period 2020–2025 (48.6%) [49,91,92,93,94,95,96,97,98,99,100,101,102,103,104,105,106,107,108,109,110,111,112,113,114,115,116,117,118,119,120]. Around two-thirds of all episodes were from Asian countries (55 out of 70 cases, 78.6%) (Appendix B, Table A1), with most of them from India (28 cases, 40.0%) and Bangladesh (7 cases, 10.0%). A total of 12 cases were reported from South America (17.1%), most of them from Brazil (7.1%) and Peru (4.3%), with 1 case each from Argentina, Colombia, Ecuador, and Panama. One case each was eventually reported from North America (United States), Oceania (French territory of New Caledonia), and Europe (imported case to mainland France).

The quality of the retrieved papers was, in most cases, high or even very high (Table 2), as the main texts provided detailed data on clinical features, diagnostic procedures, and follow-up (see also Appendix B, Table A3). Only the reports from Krithika et al. [77], Karoli et al. [90], Kodisinghe et al. [67], Prabhu et al. [119], and Chen et al. [63] appeared to be lacking information on data reporting and/or accurate follow-up, but they were ultimately included in the analyses in order to gather the largest base of evidence available.

### 3.2. Demographics

Most of the retrieved cases occurred in males (62.9%), and 80.0% were reported in subjects younger than 45 years (Figure 2), with a mean age of 31.6 years ± 14.5. Around one-quarter of cases complained of previous comorbidities (27.1%), including diabetes (7.1%), obesity (4.3%), and a previous history of hypertension (2.0%). When focusing on other commonly acknowledged risk factors for acute pancreatitis, a history of alcohol consumption was only reported in two cases (2.9%), while no one among the reported cases had a history of acute or chronic cholecystitis, including a previous diagnosis of gallstones (Appendix B, Table A3).

### 3.3. Diagnosis of Dengue Fever

Most cases were then diagnosed by means of ELISA-based testing for IgM (57.1%) and/or IgG (17.1%), and/or NS1-based antigen tests (50.0%). In only one case (1.4%) [119], laboratory work-up for dengue was performed and proved DENV infection, but details were not provided in the full details. Finally, the early case from Alvarez et al. [59] was diagnosed by means of a hemagglutination test (1.4%), as, at that time, neither ELISA nor NS1-based antigen tests were available. In seven cases, only the diagnosis of DENV infection was documented by RT-qPCR (10.0%), but most of the reported cases occurred in areas characterized by high socio-economic deprivation, where molecular tests were not available (Appendix B, Table A3). Therefore, although only 2 cases were formally classified as secondary dengue by the authors [67,118], a total of 14 cases could more plausibly be considered secondary infections [58,61,62,63,66,67,77,80,83,87,98,110,116,118], likely representing an underestimation.

### 3.4. Clinical Features

High fever was consistently reported in the days before the hospital admission (5.7 days ± 7.8 since the onset of the fever). At admission, nearly half of the participants were affected by tachycardia (50.0%) and/or hypotension (44.3%), and 34.3% of cases were characterized by both conditions (Appendix B, Table A3). Most of the cases reported epigastric pain (84.3%), while other common features were fever (body temperature ≥ 38.0 °C; 65.7%), vomiting and nausea (58.6% and 54.3%, respectively), body aches (48.6%), and headache (40.0%), while a more limited proportion of cases complained of retro-orbital pain (24.3%) or was affected by signs of altered consciousness (24.3%), skin rash (22.9%), jaundice (10.0%), and conjunctivitis (7.1%). Around one-third of all cases showed some degree of dyspnea and/or tachypnea (30.0%), while signs more specifically associated with dengue fever, such as minor bleedings (15.7%), capillary refill disorders, and peripheral edemas, were reported by a more limited share of patients (12.9% for both conditions).

### 3.5. Imaging

Imaging data were available for 67 cases out of 70 (95.7%), including both ultrasonographic studies (68.7%) and CECT studies (67.2%) (Appendix B, Table A3). Overall, 29 cases (41.4%) were studied with both ultrasonographic and CECT studies, while 19 cases (27.1%) were studied only through ultrasonographic studies, and 18 by means of CECT scans (25.7%). Imaging studies were performed by means of ^99m^Tc sulfur colloid scan only in the oldest report from Alvarez et al. [59]. The most frequently reported feature was a swollen pancreas (88.6%), while in 55.7% of cases, intrinsic pancreatic abnormalities, including areas of necrosis, were noticed. Inflammatory changes in peripancreatic fat were observed in 37.1% of cases, with signs of necrosis in 15.7%. Intrapancreatic or peripancreatic fluid collections were detected in 31.4% of cases. Ascites and pleural effusions were identified in 57.1% and 38.6% of cases, respectively, with 22.9% showing both pleural effusions and ascites. Fluid collections were reported in 29 cases (41.4%), and the most common locations were peripancreatic (62.1% of all collections), followed by intrapancreatic (27.6%), epigastric (13.8%), perihepatic, and within the lesser omental sac (10.3% for both), as well as the splenic hilum (6.9%). A single case of collection was noted in the mesenteric, paracolic, and pericholecystic regions (3.4%). Finally, two cases of collection were reported from the psoas muscle (6.9%). Overall, seven cases of collection were surgically drained (24.1%).

### 3.6. Laboratory Exams

Notably, the retrieved studies were affected by some inconsistencies in the laboratory work-up (Appendix B, Table A4 and Table A5). While most reports included data on blood cell counts, including thrombocytes (62 out of 70, 88.6%), white blood cell counts (75.7%), and hemoglobin (68.6%), as well as on enzymes associated with pancreatic functions (total lipase, 78.6%; total amylase, 87.1%), data on transaminases (ALT, 72.9%; AST, 64.4%), creatinine (50.0%), alkaline phosphatase (40.0%), bilirubin (38.6%), albumin (20.0%), GGT (14.3%), and even glycemia upon admission (32.9%) were not consistently reported.

A detailed report on the retrieved laboratory exams is provided in Appendix B, Figure A3 and Figure A4. Briefly, the available studies suggested the extensive involvement of pancreatic and liver cells, with increased levels of transaminases (AST and ALT exceeding normal ranges in 95.6% and 80.4% of cases, respectively), serum amylase and lipase (91.8% and 90.9%, respectively), and bilirubin (81.5%). The gamma-glutamyl transferase levels were above normal ranges in most of the reported cases (90.0%), but only 10 patients were specifically sampled. Nonetheless, hyperglycemia was reported in 14 out of 22 sampled cases (60.9%). Renal involvement was eventually suggested by increased levels of creatinine in 15 out of 35 sampled cases (42.9%).

Regarding blood cell counts, most of the sampled cases experienced some degree of thrombocytopenia (<150,000 platelets/µL; 83.9%). White blood cell (WBC) counts were characterized as normal in 39.6% of cases, while signs of leukopenia and leukocytosis were reported by 34.0% and 26.4% of cases, respectively. Similarly, hemoglobin levels were normal by age and gender in 28.9% of cases and were reduced (i.e., anemia) in 57.8% of cases, and signs of hemoconcentration were reported by 13.3% of cases.

### 3.7. Diagnosis and Characteristics of Pancreatitis

Most cases were characterized as AIP (65.7%), and the remaining 24 cases were characterized as ANP (34.4%). The computed tomography severity index was either provided or calculated for 45 out of 47 case reports providing computed tomography scans (i.e., 64.3% of the total sample). A mean of 4.6 ± 2.6 was calculated (range: 0 to 10) (Appendix B, Table A3). Where both ultrasonographic appraisal and CECT scans were available (30 cases, 42.9% of total), diagnostic concordance was reported for 16 cases, with 1 false-negative case in both ultrasonography and CECT scans. Twelve cases of acute pancreatitis were initially assessed as negative by abdominal ultrasonography, two of them with CTSI scores equal to eight, suggesting a substantial lack of sensitivity (56.7%).

### 3.8. Clinical Course and Outcome

Overall, 17 cases developed signs of multiorgan failure (24.3%), including 15 cases of acute kidney failure and 12 cases of respiratory failure (17.1%). Following the completion of the diagnostic work-up, 51.4% cases were classified as mild cases, while 21.4% were considered moderately severe and 27.1% were considered severe. The mean hospital stay was estimated as 12.2 days ± 8.8, ranging from 3 to 51 days, with eight deaths associated with complications of acute pancreatitis and dengue infection (11.4%) (Appendix B, Table A3).

### 3.9. Univariate Analysis

No substantial differences were reported by the gender of patients (see Appendix B, Table A3 for details). Similarly, AIP was more frequently identified among younger subjects, and ANP was more frequently identified among older individuals, but the eventual difference was not significant (*p* = 0.467). No substantial differences were identified in the distribution of cases by their severity in the various age groups (*p* = 0.946 for severe cases vs. mild and moderately severe cases; Figure 2).

Regarding the clinical features of pooled cases (Appendix B, Table A3), ANP was associated with a higher frequency of collections compared to AIP (75.0% vs. 23.9%, *p* < 0.001), while ANP cases were more frequently associated with signs of multiorgan (45.8% vs. 13.0%, *p* = 0.002) and respiratory (33.3% vs. 8.8%) failure than AIP. In fact, ANP cases were more frequently characterized as either severe (62.5%) or moderately severe (25.0%) than AIP cases (8.7% and 19.6%, respectively), which were mostly characterized as mild. Similarly, the hospital stay was substantially longer among patients affected by ANP than among those affected by AIP (16.6 ± 12.7 vs. 10.1 ± 5.0, *p* = 0.030), and the case fatality ratio of ANP was more than five times higher than that reported for AIP (25.0% vs. 4.3%, *p* = 0.029).

Focusing on the laboratory work-up based on reference levels, ANP was only characterized by higher WBC levels, as the leukocytosis range was identified in more than half of 17 ANP patients reporting WBC counts (52.9%), compared to 13.9% of AIP cases (*p* = 0.007; Appendix B, Table A4). As shown in Appendix B, Table A5, the WBC count at admission (11,953.5 cells/µL ± 9445.9 vs. 6383.6 cells/µL ± 4616.9, *p* = 0.039) and serum amylase levels (1300.6 U/L ± 791.6 vs. 609.4 U/L ± 514.3, *p* = 0.001) were significantly higher in ANP compared to AIP cases. When the receiver operating characteristic curve (ROC curve) was plotted for serum amylase and WBCs on the diagnosis of ANP vs. AIP, a distinctive pattern was similarly identified (Appendix B, Figure A5). Corresponding areas under the curve (AUC) were estimated: 0.776 (95%CI: 0.637 to 0.914, *p* = 0.001) for amylase and 0.678 (95%CI: 0.505 to 0.852, *p* = 0.039) for WBCs. Corresponding best cut points for diagnosis of ANP vs. AIP were calculated by means of Youden’s J statistics (i.e., J = sensitivity + specificity − 1) as >8300 cells/µL for WBCs and >740 U/L for amylase.

Moreover, acute pancreatitis-associated deaths were more frequently reported among patients characterized by higher serum levels of creatinine (3.7 mg/dL ± 2.6 vs. 1.5 mg/dL ± 1.4, *p* = 0.05; Appendix B, Table A5), diagnosis of ANP (i.e., 75.0% of deaths vs. 29.0% of hospital discharges, Figure 2), signs of multiorgan failure (75.0% vs. 17.7%, *p* = 0.002), respiratory failure (75.0% vs. 9.7%, *p* < 0.001), higher CTSIs (7.2 ± 3.3 vs. 4.3 ± 2.3, *p* = 0.016), and collections (85.5% vs. 35.5%, *p* = 0.015). Similarly, a dismal prognosis was significantly associated with reporting any pre-existing comorbidity (62.5% vs. 22.6%, *p* = 0.049), including obesity (25.0% vs. 1.6%, *p* = 0.032), and being affected by hypotension at the time of hospitalization (87.5% vs. 38.7%, *p* = 0.025).

### 3.10. Survival Analysis

As shown in Figure 3, not only did cases of AIP exhibit a better survival rate, but the cumulative survival at one year was also notably higher (i.e., 349.5 days; 95%CI: 328.5 to 370.5) compared to the estimates for ANP cases (275.8 days; 95%CI: 214.0 to 337.7; *p* = 0.009).

### 3.11. Multivariable Analysis

In the multivariable analysis (Figure 4, Appendix B, Table A6), Cox regression model I (i.e., diagnosis of ANP vs. AIP) included the following explanatory variables: any comorbidity, any sign of collection in imaging studies, signs of organ failure, the WBC range, and the outcome (survival vs. death). Reporting any sign of collection (HR: 5.544; 95%CI: 1.335 to 23.024) and having a WBC count in the leukocytosis range (HR: 7.990; 95%CI: 1.865 to 34.237) were significantly associated with ANP (Figure 4a), while any comorbidity (HR: 0.360; 95%CI: 0.081 to 1.602), a WBC count in the leukopenia range (HR: 1.638; 95%CI: 0.354 to 7.581), any sign of organ failure (HR: 2.538; 95%CI: 0.820 to 7.853), and death (HR: 1.418; 95%CI: 0.275 to 7.313) were not.

Model II (i.e., outcome variable represented by eventual death following acute pancreatitis, Figure 4b) included the following explanatory variables: any comorbidity, any sign of hypotension, the classification of the pancreatitis (i.e., AIP vs. ANP), any evidence of collections, and any sign of organ failure. Eventually, the analysis identified as predictive towards the eventual death of the patient the evidence of collections (HR: 15.531; 95%CI: 1.102 to 218.816) and any signs of organ failure during the hospital stay (HR: 9.030; 95%CI: 1.472 to 55.405). On the contrary, reporting any comorbidity (HR: 1.218; 95%CI: 0.175 to 8.472), signs of hypotension (HR: 7.216; 95%CI: 0.664 to 78.416), and the classification of the pancreatitis (HR: 0.820; 95%CI: 0.107 to 6.283) were not.

## 4. Discussion

### 4.1. Summary of Main Findings

Our systematic review retrieved and summarized clinical, laboratory, and imaging data from a total of 70 case reports published since 1985, 55 of them from Asian countries (78.6%). Most cases were characterized as AIP (n. 46, 65.7%), while 34.3% of cases were consistent with the diagnosis of ANP. Most reports were of mild to moderate severity (CTSI < 7), with a case fatality ratio of 11.4%, benefiting from a conservative approach that hinted at the potential significance of early enteral nutrition [49,121,122]. Most of the deaths occurred among ANP cases (n = 6), with only two deaths among AIP (chi-squared test *p*-value = 0.029). In the Cox regression model, ANP was strongly associated with leukocytosis (HR: 7.990; 95%CI: 1.865 to 32.273) and reporting any sign of collection (HR: 5.544; 95%CI: 1.335 to 23.024). Death following dengue-associated acute pancreatitis was associated with reporting any sign of collection in imaging studies (HR: 15.531; 95%CI: 1.102 to 218.816) and any sign of organ failure (HR: 9.030; 95%CI: 1.472 to 55.405).

### 4.2. Interpretation of Main Findings

According to available estimates, acute pancreatitis is a quite common cause of hospitalization, and its incidence has increased during the last 50 years [32], currently ranging from 13 to 49 cases per 100,000 persons [31,32,52]. The global incidence peaks are associated with Eastern Europe (71.2 per 100,000/year; 95%CI: 60.8 to 82.9), followed by North America (62.4 per 100,000/years; 95%CI: 53.7 to 72.0) and Andean areas of South America (45.8 per 100,000/year; 95%CI: 39.6 to 52.9) [31]. The geographic areas most extensively represented in our pooled sample (i.e., Asia and non-Andean areas from South America) are conversely associated with lower annual incidence rates (33.7 per 100,000, 95%CI: 28.4 to 39.4; 38.1 per 100,000, 95%CI: 31.6 to 45.9; 26.0 per 100,000, 95%CI: 21.6 to 30.9; 20.1 per 100,000, 95%CI: 17.7 to 22.7 for Central Asia, Southern Asia, Southeastern Asia, and Tropical Latin America) [31,32].

Gallstones and alcohol are among the most common causes of acute pancreatitis [52], accounting for up to two-thirds of all cases. However, there is some consistent evidence for the potential role of some microbial pathogens, including viruses, bacteria, parasites and fungi [33]. More precisely, in their recent systematic review, Iman et al. [36] were able to retrieve data on 320 patients with documented acute pancreatitis attributed to a microorganism, most of them being associated with viral infection (65.3%). Still, acute pancreatitis was documented as a rare complication of DENV infection [33,36,37], and its occurrence was overshadowed by more frequently reported pathogens, such as hepatotropic viruses (i.e., Hepatitis A, B, C, D, E viruses; 34.4% of total cases) [61,63,65,66,71,73,80,83,90], followed by Coxsackie viruses and echoviruses (14.8%), hemorrhagic fever viruses (12.4%), cytomegalovirus (12.0%), and even varicella zoster virus (10.5%) [33,34,123,124]. Also, the retrospective study from Mohan et al. [125] documented signs and symptoms of acute pancreatitis in only 0.4% of 464 cases of dengue, and a similar occurrence was reported by Dinkar and Singh (0.65% of 461 cases) [126]. However, most of the dengue-associated acute pancreatitis cases may likely be misdiagnosed due to the extensive overlapping with other arbovirus infection of signs and symptoms, such as high fever, hypotension, coagulation disorders, and abdominal pain [17,47,78,100,106,127,128,129]. For example, in their report on 100 laboratory-documented dengue cases, Ghweil et al. [130] identified acute pancreatitis as a late complication of DENV infection in 13% of all cases, with similar estimates from Mohanty et al. [131] (13.9%), Mahjumdar et al. [132] (15%), and Ramos-De La Medina et al. [133] (14%). An increased and previously overlooked risk for acute pancreatitis among cases of DENV infection was recently identified by Shih et al. [127] among 65,694 cases of dengue (adjusted HR: 17.13; 95%CI: 7.66 to 32.29 for dengue vs. non dengue cases). Unfortunately, the authors did not provide the proportion of AIP vs. ANP, and this specific point may be of particular interest due to the very high share of ANP in our pooled sample compared to the currently available international figures [34,38,39,40,41,52].

While the association of DENV infection with acute pancreatitis can therefore be considered unusual but documented, its etiopathogenesis remains unclear [35,71,73,77]. Some theories have been raised, including [60,83,108] the direct viral invasion of pancreatic acinar cells followed by apoptosis and necrosis, viral-induced enzyme activation with immune-mediated inflammatory damage, microvascular ischemia due to the increased vascular permeability, and even self-digestion of pancreatic tissue. In fact, there is some evidence that DENV infection ultimately elicits production of cytokines such as tumor necrosis factor alpha, transforming growth factor beta, and interleukin 10, otherwise observed in studies on non-DENV-associated acute pancreatitis [50,53,134]. In other words, while the ability of DENV to directly infect pancreatic parenchymal cells is far from proven, the DENV-infection-driven production of cytokines and inflammatory mediators may contribute to the extensive interstitial damage of the gland. As ANP is, in turn, associated with immune-mediated damage and/or ischemic injury from increased vascular permeability [135], this speculative explanation could explain the very high proportion of ANP we otherwise identified [64,96,105,108].

### 4.3. Generalizability of Main Findings and Implications for Daily Practice

While the principal findings of our study may carry substantial implications, several potential issues warrant consideration in terms of the potential implications for daily practice. First of all, given the ongoing debate regarding the pathogenesis of acute pancreatitis in the context of DENV infection—and considering that both acute pancreatitis and DENV infection are relatively common—the incidental diachronous or synchronous occurrence of these conditions should be carefully evaluated, particularly in case reports originating from regions with high or very high endemic viral circulation [37,74,106,126,133,136,137]. Interestingly enough, common risk factors for acute pancreatitis, such as biliary tract obstruction due to gallstones and alcohol intake, medical procedures including retrograde cholangiopancreatography, medications including furosemide and losartan, metabolic disorders such as hypercalcemia, and surgery and malignant conditions affecting the biliary tract, have been ruled out for the whole of our sample [33,34,36,39,49,50,51,52,53,122,135]. Only two cases (2.9%) had any history of alcohol consumption, while neither medical history nor imaging studies identified any sign of gallstones. Hence, professionals involved in the management of DENV infections should be aware that acute pancreatitis should be suspected in cases of dengue where the epigastric pain is associated with a documented and significant increase (i.e., threefold increase) [48,138] in serum levels of lipases and amylase [60,83], even without any other documented risk factor.

Second, in most cases, hospitalization for acute pancreatitis symptoms occurred days to weeks after the onset of DENV infection, as suggested by laboratory diagnosis and/or the reported evidence of episodes of high fever (5.7 days ± 7.8) with subsequent laboratory testing for DENV. As an appropriate laboratory work-up for other more common viral pathogens (e.g., mumps virus, coxsackievirus, cytomegalovirus, varicella zoster virus, herpes simplex virus, and even human immunodeficiency virus) [33,34,36,41,48,49,50,51,53,90,127] was not consistently performed, we cannot rule out that some of the cases we included in our pooled sample may have been complicated by other viral infections.

Third, as most of the pooled cases occurred in the first weeks following DENV infection, acute pancreatitis seemingly represented a relatively late complication of primary infection [124,127]. The delay between the diagnosis of DENV infection and the onset of signs and symptoms may have therefore complicated the identification of the causal link, while it could potentially contribute to the underestimation of DENV-associated acute pancreatitis cases [127,128], particularly for AIP (i.e., clinical conditions that mostly benefit from a benign outcome through supportive approach), particularly during seasonal outbreaks, potentially leading to the oversampling of more severe ANP cases.

Fourth, most of the signs and symptoms of acute pancreatitis and DENV infection may be misunderstood due to their substantial overlap, and relying on laboratory exams, and particularly on lipase and amylase levels, could lead to substantial inaccuracy. Although increased levels of both markers were documented across nearly all the retrieved and pooled dengue-associated acute pancreatitis cases, DENV infection may be associated with somewhat increased levels of serum amylase levels even without any evidence of actual pancreas involvement [130,139]. Even though amylase is mostly synthesized by pancreatic acinar cells, other organs could contribute to its production [48]. While the amount of serum amylase secreted by adipose tissue, gonads, fallopian tubes, the intestinal tract, and skeletal muscles is usually negligible, salivary glands substantially contribute to the total serum amylase levels [140,141] and can be directly involved in several clinical conditions [48,138], including DENV infection [2,141]. As most of the laboratory assays are unable to discriminate between pancreatic and salivary isoenzymes [48,138], increased levels of serum amylase could therefore be of limited specificity for the diagnosis of acute pancreatitis [48]. Moreover, elevated levels of total serum amylase could result not only from direct damage to the cells involved in its synthesis but also from the impairment of its removal from renal proximal tubules and liver cells [48,138]. Again, even the severe involvement of both kidney and liver functions during severe dengue is quite common [5,65,111,133,142]. Similar to amylase, elevated serum lipases associated with abdominal discomfort or even with acute abdominal pain may be due to other non-pancreatic conditions, such as trauma, appendicitis, ketoacidosis, intestinal obstruction, renal failure, and hypertriglyceridemia [48,138], including organ involvement due to DENV infection other than the pancreatic one [5,65,111,133,142]. Finally, the rise in serum amylase and lipase peaks at three to six hours following the onset of acute pancreatic damage, with a half-life ranging from 10 to 12 h and persistent elevation lasting up to five days [48,138]. While a cautious approach should rely on repeated measurements, at least in the first 24 h from the onset of abdominal pain, early, negative results of lipase or amylase tests could yield a false-negative appraisal of underlying acute pancreatitis [48].

While most international guidelines in recent years have shifted toward a selective preference for lipase over amylase, the small number of cases analyzed and the lack of test-negative controls preclude any evidence-based conclusions regarding the validity of this approach for the differential diagnosis of DENV-associated acute pancreatitis [48,138]. On the contrary, in the pooled cases we were able to retrieve, levels of serum amylase > 740 U/L represented a likely cut-off for dichotomizing the potential diagnosis of ANP from AIP.

In fact, both elevated lipase and/or amylase should only serve as the first steps in the diagnosis of acute pancreatitis, both in general and in the specific case of suspected DENV-associated acute pancreatitis. In this regard, our study stresses the key importance of guaranteeing proper and early access of suspected cases to medical imaging, and particularly to CECT scans, stressing how ultrasonography cannot be considered an alternative diagnostic option for the proper work-up of DENV-associated acute pancreatitis cases. In recent years, ultrasonography has sometimes been regarded as a cheaper and cost-effective alternative to CECT, particularly in low-resource settings [143,144,145]. Moreover, ultrasonography can be performed as a bedside, point-of-care option, directly at hospital admission [143,144,146], reducing the turnaround time during the clinical work-up. Unfortunately, the pooled sensitivity of ultrasound was substantially lower compared to CECT scans (56.7%), suggesting that suitable cases cannot be properly assessed without accurate imaging.

Our study also possibly shed some light on the high occurrence of ANP over the whole of dengue-associated acute pancreatitis cases we collected. While the majority of acute pancreatitis cases are usually represented by AIP [40,50,52,53,121,147], more than one-third of the retrieved case reports included ANP cases. The oversampling of ANP cases compared to AIP could be explained in terms of potential publication bias or by means of biological mechanisms, considering that the available data do not allow any definitive distinction between these two hypotheses. On the one hand, we cannot rule out the oversampling of more severe and complicated cases due to the design of our research strategy, based on case reports, with a larger proportion of AIP cases simply undiagnosed as characterized by mild signs and symptoms, or even misdiagnosed [57]. On the other hand, the high proportion of ANP may be associated with the DENV invasion of acinar cells, which are ultimately destroyed, and/or immune-mediated inflammatory damage and/or microvascular ischemia due to increased vascular permeability associated with DENV infection [64,69,71,106], possibly explaining the higher levels of serum amylase compared to AIP cases. Unfortunately, most of the collected data may be of limited help on this specific topic, as both the evidence of collections (ANP vs. AIP, HR: 5.544; 95%CI: 1.335 to 23.024) and leukocytosis range (HR: 7.990; 95%CI: 1.865 to 34.237) can be considered a consequence of underlying conditions rather than a risk factor.

Even though our study was not specifically designed for providing detailed insights on the therapeutic options of DENV-associated acute pancreatitis, some details may be of certain interest for the daily practice of clinicians managing similar cases. First of all, only seven cases required any invasive approach for collection drainage [66,74,89,98,99,104,120], suggesting that without signs of multiorgan failure, most of the cases, including severe cases characterized by ANP and collections, could be conservatively managed, guaranteeing an appropriate alimentary support and adequate volume replacement, being mostly affordable even in limited resource settings. Conversely, cases showing any need of respiratory support [35,65,71,73,80,89,92,103,113,119,120] or signs of renal failure [35,58,59,60,65,71,73,80,83,91,92,97,101,111,115] should be very cautiously monitored and could benefit from more aggressive therapies, as they are more likely characterized by severe systemic complications and a dismal prognosis.

### 4.4. Limits and Implications for Future Studies

Although the present study represents the largest synthesis to date of DENV-associated pancreatitis with detailed clinical and imaging information, several limits should be acknowledged, most of them due to the source studies and data collection. First, albeit of relatively high quality, the studies included in this dataset were either case reports or case series, and the total number of pooled subjects remains low. In fact, case reports and case series are descriptive studies illustrating novel, unusual, or atypical features identified in patients in medical practice, and substantial disagreement remains regarding their value in the medical literature, particularly when dealing with the collection of evidence [57,148].

Second, most cases were retrospectively retrieved from medical registries after the diagnosis of either pancreatitis or DENV infection. Hence, some degree of recall bias cannot easily be ruled out, being possibly complicated by the various allowances of care delivery and research resource availability across reporting hospitals. Therefore, the present dataset forcibly captured the heterogeneity due to the different diagnostic options and inter-observer inconsistencies. For instance, laboratory work-up data were inconsistently reported, as stressed by the uneven availability of all exams. Also, imaging studies were non-systematically performed by means of very heterogeneous computed tomography machines, and across regions affected by dengue where the healthcare preparedness to deal with outbreaks is strikingly variable [149,150]. Three cases lacked data on CECT scans, and CTSIs were either available or reverse-calculated from the available scans in a reduced proportion of cases (45 out of 70 cases). Moreover, as the performance of computed tomography scans may have been associated with clinicians’ suspicions for higher severity, we cannot rule out the potential overestimation of the actual severity of cases [38,55,144,151].

Third, the diagnostic options for DENV infection were quite heterogeneous, including RT-qPCR, the antigen detection of NS1, and ELISA with detection of DENV-IgM and DENV-IgG. The choice of diagnostic option, as often stressed by most studies [58,61,74,79,80,89,98,100,103,105,110,111,115,136], was more frequently due to economic constraints than to medical choices relying on diagnostic performances [152]. For instance, according to a recent systematic review with meta-analysis [152], the IgM ELISA is characterized by poor diagnostic accuracy (sensitivity: 62%, 95%CI: 45 to 75; specificity: 85%, 95%CI: 76 to 91), particularly during the early stages of the syndrome (sensitivity: 17%, 95%CI: 3 to 51; specificity: 84%, 95%CI: 47 to 97 for days 0 to 4). On the contrary, NS1 testing usually showed high diagnostic accuracy, comparable to RT-qPCR, in both the early stages of the symptomatic infection (pooled sensitivity for days 0 to 4 from the onset of symptoms: 90%, 95%CI: 68 to 98, and 95%, 95%CI: 77 to 99 for NS1 testing and RT-qPCR, respectively; pooled specificities of 93%, 95%CI: 71 to 99, and 89%, 95%CI: 60 to 98, for NS1 testing and RT-qPCR, respectively) and in the whole of symptomatic subjects (pooled sensitivities of 85%, 95%CI: 76 to 91, and 86%, 95%CI: 63 to 93, for NS1 testing and RT-qPCR, respectively; pooled specificities of 90%, 95%CI: 83 to 94, and 96%, 95%CI: 84 to 98, for NS1 testing and RT-qPCR, respectively). As more than half of the included cases were diagnosed via IgM ELISA, the actual diagnosis of the included cases should be carefully addressed. Subject to the financial resources of the healthcare systems involved, a more rigorous and systematic use of diagnostic panels incorporating both serological assays and molecular/NS1 testing is therefore desirable.

In this regard, the available studies did not provide consistent information about the DENV genotype associated with reported cases. Despite their similarities, DENV-1, -2, -3, and -4 are associated with a substantial degree of genetic difference [2,10,15], which has been previously understood as a potential cause of the antibody-dependent enhancement, the most feared complication of DENV infection [2,13,14,15,20,153]. In turn, within each serotype, genetic differences have been reported, possibly impacting epidemiology and clinical features [2]. Therefore, by lacking this specific information, we cannot rule out that the high occurrence of ANP and the heterogeneous prognosis we documented among recollected cases may be associated with some specific serotype or DENV strain. Future reports could fill this significant information gap by providing more accurate serotype or genotype information.

Fourth, the retrospective nature of this dataset resulted in its reasonably heterogeneous definition in terms of the clinical stage of underlying DENV infection and acute pancreatitis. In other words, we cannot rule out that both reported clinical features and potential risk factors for acute pancreatitis (and for AIP and ANP) should be reconciled with the clinical stage of pancreatitis and dengue infection, rather than with individual characteristics of that specific case. This is specifically relevant when dealing with the recollection of signs and symptoms, as their evolution over time during the clinical course of dengue has been accurately described [1,5,10,16,17,20,21].

Fifth, only two studies included a formal definition of reported cases as either primary or secondary dengue [67,118]. Even though the analysis of the available serology suggests that a total of 14 cases could plausibly be considered secondary infections [58,61,62,63,66,67,77,80,83,87,98,110,116,118], this estimate likely represents an underestimation due to the inconsistent serology from several studies. As a consequence, the lack of association of secondary status with the more severe features usually associated with ANP (20.8% vs. 19.6% of included AIP cases, *p* = 1.000) and the eventual death of the patient due to DENV-associated acute pancreatitis (all secondary cases had a generally favorable clinical outcome, *p* = 0.302; Appendix B, Table A3) should be quite cautiously appraised.

In other words, the actual association of individual risk factors and clinical features with DENV-infection-associated acute pancreatitis should be very cautiously assessed, as suggested in some previous studies [37,107,124,125,128,130,132,154], and a precautionary approach is needed.

## 5. Conclusions

Acute pancreatitis associated with DENV infection has been increasingly reported, suggesting that pancreatitis may represent a true complication of dengue rather than a serendipitous association. It should be suspected in dengue patients presenting with epigastric pain and significant enzyme elevations; contrast-enhanced CT is important for identifying ANP and collections. Even though the available evidence is limited, acute pancreatitis associated with dengue infection is characterized by a relatively high case fatality rate and the higher occurrence of ANP compared to AIP. DENV-associated acute pancreatitis should be acknowledged as a potentially severe condition, with mortality driven by collections and organ failure. Nonetheless, the substantial overestimation of its actual severity cannot be ruled out. Due to the increasing global burden of DENV infection and the significance from both clinical and public health points of view of acute pancreatitis as a potential complication, our data stress the importance of a more appropriate definition of the actual epidemiology of this condition by means of studies specifically targeting the occurrence of acute pancreatitis during the clinical course of DENV infection.

## Figures and Tables

**Figure 1 tropicalmed-10-00330-f001:**
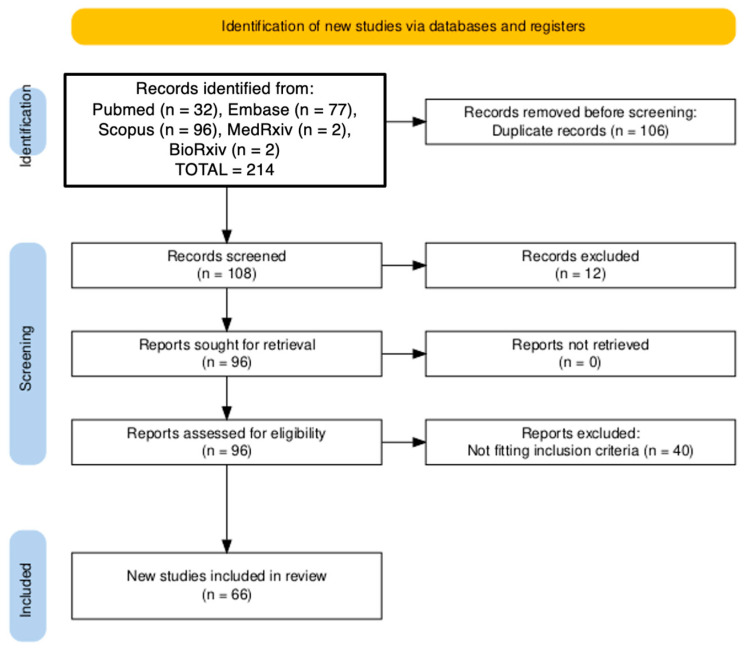
PRISMA (i.e., Preferred Reporting Items for Systematic Reviews and Meta-Analyses) flowchart of included studies [42,43].

**Figure 2 tropicalmed-10-00330-f002:**
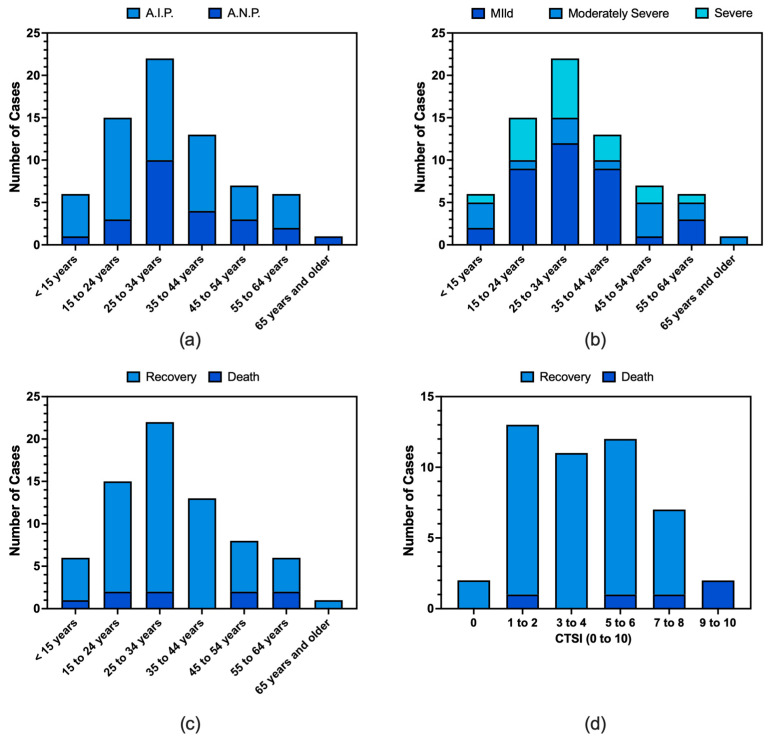
Distribution of cases of pancreatitis by subtype (subfigure **a**) and severity (subfigure **b**), and their outcomes by age groups (subfigure **c**) and computed tomography severity index (CTSI) (potential range: 0–10) (subfigure **d**). No substantial differences were eventually identified for pancreatitis subtype (**a**), severity considering severe cases vs. mild and moderately severe cases (**b**), or regarding the distribution of deaths by age groups (**c**) (chi-squared test *p*-value 0.467, *p* = 0.946, *p* = 0.538, respectively). On the contrary, deaths were clustered in cases characterized by higher CTSI scores (chi-square = 18.699; df = 5, *p* = 0.005).

**Figure 3 tropicalmed-10-00330-f003:**
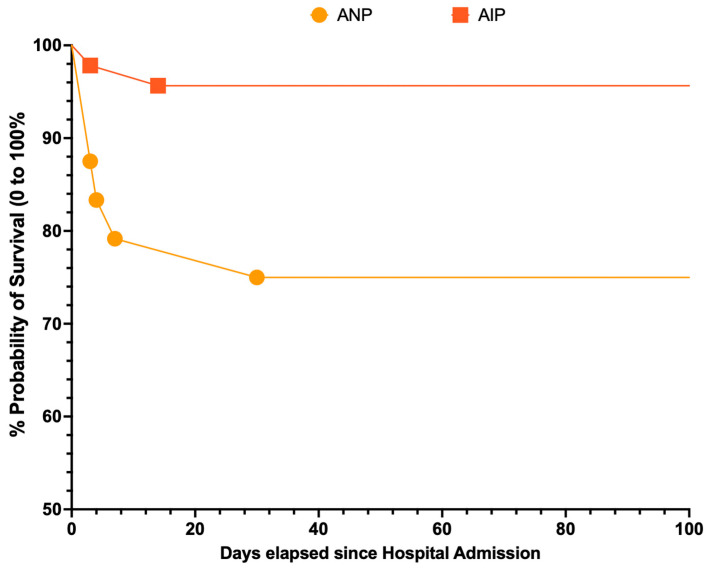
Survival analysis of retrieved cases of acute pancreatitis during acute dengue virus infection by subtype. Acute necrotizing pancreatitis (ANP) cases exhibited a more severe outcome than that of acute interstitial pancreatitis (AIP) cases (Mantel–Cox test chi-square = 6.772; *p* = 0.009).

**Figure 4 tropicalmed-10-00330-f004:**
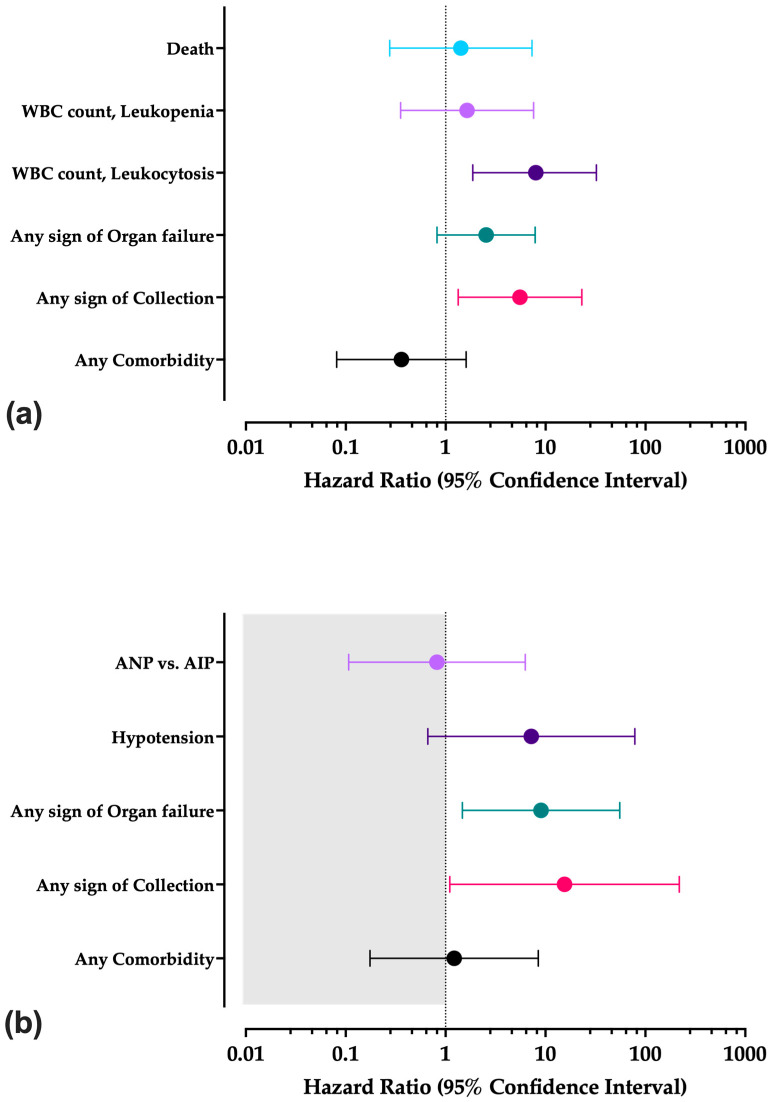
Results of Cox regression models with calculation of corresponding hazard ratios with their 95% Confidence Intervals for acute necrotizing pancreatitis (ANP) vs. acute interstitial pancreatitis (AIP) (subfigure **a**), and for the eventual outcome of the episode (subfigure **b**), from 70 cases with documented dengue virus infection.

**Table 1 tropicalmed-10-00330-t001:** Summary of search strategies across search databases.

Database	Search Strategy	Entries
PubMed	(“Dengue”[Mesh] OR “Severe Dengue”[Mesh] OR “Dengue Virus”[Mesh]) AND (“Pancreatitis”[Mesh] OR “pancreatitis” OR “acute pancreatitis” OR “necrotizing pancreatitis” OR “interstitial pancreatitis” OR “hemorrhagic pancreatitis”)	32
EMBASE	(‘dengue’/exp OR ‘dengue’ OR ‘dengue virus’) AND (‘acute pancreatitis’ OR ‘acute hemorrhagic pancreatitis’ OR ‘acute interstitial pancreatitis’)	77
Scopus	((acute) OR (interstitial) OR (necrotizing)) AND (Pancreatitis) AND ((Dengue) OR (Severe Dengue) OR (Dengue Virus))	96
MedRxiv		2
BioRxiv		7

**Table 2 tropicalmed-10-00330-t002:** Summary of 70 cases of acute pancreatitis diagnosed during episodes of dengue virus infection.

Reference	Year	Country	Age (years)	Gender	Day ^1^	Additional Diagnoses	Dengue Diagnosis	Severity	Class.	CT Score (0–10)	Outcome	Quality ^2^ (0–6)
Acherjya et al., 2023 [100]	2022	Bangladesh	55	Male	3	-	NS1	M	Int.	4	Survived	6
Agrawal et al., 2011 [73]	2011	India	38	Male	10	-	IgM	S	Nec.	6	Survived	6
Ahmed et al., 2024 [115]	2024	Pakistan	20	Male	1	-	NS1	S	N.A.	N.A.	Survived	4
Alnuaimi et al., 2025 [108]	2024	United Arab Emirates	44	Female	5	-	PCR	M	Int.	2	Survived	6
Alvarez et al., 1985 [59]	1981	United States	51	Male	1		HA	MS	Nec.	-	Survived	3
Anam et al., 2016 [80]	2013	Bangladesh	20	Male	8	-	IgM, IgG	S	Int.	2	Survived	5
Anam et al., 2017 [79]	2015	Bangladesh	44	Male	1	-	NS1	M	Int.	N.A.	Survived	6
Arora et al., 2023 [92]	2023	India	47	Male	3	-	NS1, IgM	MS	Int.	4	Survived	4
	2023	India	63	Female	5	DM, HPT, OC	IgM	MS	Nec.	6	Survived	5
	2023	India	45	Male	6	HPT	IgM	S	Nec.	8	Death	5
Arredondo-Nontol et al., 2022 [113]	2022	Peru	13	Male	3	OBS	NS1	MS	Int.	4	Survived	6
Bashir Bhatti et al., 2015 [81]	2015	India	16	Female	3	-	NS1	M	Int.	1	Survived	6
Beaussac et al., 2015 [49]	2020	France (European)	18	Female	7	-	IgM	M	Int.	0	Survived	3
Biswas et al., 2024 [103]	2024	India	35	Female	5	ENC, HEP	IgM	S	Int.	4	Survived	6
Che Jusoh et al., 2016 [88]	2015	Brunei	22	Male	3	-	PCR, NS1	MS	Int.	4	Survived	5
Chen et al., 2004 [63]	2004	Taiwan	66	Female		DM, seizures	IgM, IgG	MS	Nec.	6	Survived	3
Correa et al., 2019 [84]	2018	Panama	37	Female	5	-	PCR, IgM	M	Int.	-	Survived	6
Dalugama et al., 2017 [87]	2016	Sri Lanka	26	Male	3	-	NS1, IgM, IgG	M	Nec.	-	Survived	5
de Andrade et al., 2022 [35]	2002	Brazil	63	Male	1	DM	IgM	MS	Nec.	6	Death	6
	2002	Brazil	21	Female	8	OBS	IgM	S	Nec.	-	Death	4
Derycke et al., 2005 [62]	2005	France (Nouvelle Caledonie)	29	Male	1	-	PCR, IgM, IgG	MS	Int.	2	Survived	6
Devi et al., 2022 [76]	2019	Pakistan	35	Male	3	-	IgM	M	Int.	-	Survived	4
dos Passos Cunha et al., 2021 [89]	2019	Brazil	58	Male	3	TB, cancer (thymoma), pleuropneumonectomy	PCR	S	Int.	2	Death	6
Duarte et al., 2024 [101]	2023	Argentina	28	Male	14	Previous removal of gallbladder	NS1	M	Int.	-	Survived	4
Dutta et al., 2025 [93]	2025	India	4	Male	1		NS1	S	Nec.	6	Survived	6
Flor et al., 2022 [110]	2021	Ecuador	26	Male	5	Alcohol (abuse)	IgM, IgG	M	Int.	-	Survived	6
Gomes et al., 2021 [94]	2021	Bangladesh	35	Female	4	-	NS1	M	Int.	-	Survived	6
Gonzalez-Fontal et al., 2011 [64]	2009	Colombia	27	Male	4	ERSD, DM	IgM	M	Int.	4	Survived	6
Iqbal et al., 2012 [82]	2012	India	40	Female	4	Scrub typhus	IgM	M	Int.	4	Survived	6
Islam et al., 2023 [104]	2023	India	25	Male	4	-	NS1	M	Nec.	-	Survived	6
Jahan et al., 2020 [118]	2020	Bangladesh	42	Male	3	-	IgM	M	Int.	-	Survived	6
Jain et al., 2014 [71]	2014	India	27	Male	4	Sickle cell	NS1	S	Nec.	N.A.	Death	6
Jusuf et al., 1998 [58]	1997	Indonesia	25	Female	5	-	IgM, IgG	S	Nec.	8	Survived	4
Karoli et al., 2012 [90]	2012	India	35	Female		-	NS1	MS	Nec.	-	Survived	3
Kashyap et al., 2024 [105]	2024	Nepal	24	Female	4	-	IgM	M	Int.	2	Survived	6
Khanal et al., 2023 [99]	2022	Nepal	25	Male		-	NS1	M	Int.	-	Survived	5
	2022	Nepal	51	Male		-	NS1	M	Nec.	6	Survived	4
Khataniar et al., 2023 [98]	2023	India	28	Female	2	-	IgM, IgG	S	Nec.	8	Survived	6
Kodisinghe et al., 2011 [67]	2011	Sri Lanka	35	Male	4	-	IgM	M	Int.	-	Survived	3
Krithika et al., 2018 [77]	2017	India	11	Male	7	-	IgM, IgG	M	Int.	4	Survived	2
Kumar Das et al., 2024 [117]	2023	India	26	Male	3	-	NS1	MS	Int.	6	Survived	6
Kumar et al., 2016 [68]	2016	India	10	Female	1	AIHA	NS1, IgM	MS	Int.	-	Survived	6
Kumar et al., 2017 [70]	2017	India	32	Female	4	-	IgM	M	Int.	-	Survived	6
Kumar et al., 2018 [69]	2018	India	8	Female	3	-	NS1, IgG	M	Int.	2	Survived	6
Lee et al., 2013 [66]	2013	Taiwan	47	Male	6	HBV	IgM, IgG	S	Nec.	8	Survived	6
Lee et al., 2021 [120]	2021	Malaysia	31	Female	10	psoas muscle hematoma, cholestatic HEP	NS1	MS	Nec.	6	Survived	4
Lu et al., 2020 [85]	2019	China	33	Female	5	-	PCR, NS1, IgM	M	Int.	2	Survived	6
Mishra et al., 2019 [75]	2019	India	17	Female	5	-	IgM	S	Nec.	10	Death	6
Naik et al., 2021 [106]	2021	India	21	Male	3	-	NS1	M	Int.	2	Survived	6
Nakazaki et al., 2024 [109]	2023	Peru	18	Female	5	-	NS1	M	Int.	4	Survived	6
Nawal et al., 2018 [72]	2016	India	25	Male	3	-	NS1	M	Int.	4	Survived	6
Nguyen et al., 2023 [114]	2023	Vietnam	31	Male	2	-	NS1	M	Int.	8	Survived	5
Nogueira et al., 2022 [91]	2022	Brazil	41	Male	2	Previous pancreatitis (EBV)	NS1, IgM	S	Nec.	8	Survived	6
Prabhu et al., 2025 [119]	2025	India	34	Male	60	Kidney transplant, CRD	N.P.	S	Nec.	10	Death	3
Rahman et al., 2020 [97]	2021	Bangladesh	32	Female	2	-	NS1	M	Int.	-	Survived	6
Rajesh et al., 2008 [34]	2008	India	16	Female	21	Cassava	IgM	M	Int.	2	Survived	3
Ramindla et al., 2025 [95]	2025	India	17	Male	10	-	IgM	M	Int.	6	Survived	6
Ridho et al., 2000 [60]	2000	Indonesia	28	Male	2	-	IgM	M	Int.	-	Survived	6
Rodriguez Gonzalez et al., 2025 [116]	2025	Peru	17	Female	4	-	NS1, IgM, IgG	M	Int.	2	Survived	6
Saber et al., 2021 [102]	2021	Bangladesh	47	Male	4	-	NS1	MS	Int.	-	Survived	6
Seetharam et al., 2010 [78]	2010	India	56	Male	1	-	IgM	M	Int.	2	Survived	4
Sharma et al., 2018 [74]	2018	India	32	Male	4	-	NS1	S	Nec.	6	Survived	6
Simadibrata et al., 2012 [83]	2012	Indonesia	59	Male	4	-	IgM, IgG	M	Int.	4	Survived	6
Singh Lakra et al., 2022 [96]	2022	India	15	Male	5	-	NS1, IgM	S	Nec.	8	Survived	5
Sudulagunta et al., 2016 [65]	2016	India	30	Male	6	DM	PCR, NS1, IgM	S	Nec.	6	Survived	6
Teja Derbesula et al., 2016 [86]	2016	India	36	Male	20	-	NS1	M	Nec.	6	Survived	6
Thadchanamoorthy et al., 2022 [111]	2022	Sri Lanka	6	Female	5	-	NS1	MS	Int.	-	Death	5
Ullah et al., 2025 [107]	2024	Pakistan	25	Male	14	-	NS1	M	Int.	-	Survived	5
Vilela Assis et al., 2021 [112]	2020	Brazil	24	Female	2	-	IgM	M	Int.	0	Survived	6
Wijekoon et al., 2010 [61]	2009	Sri Lanka	47	Male	7	Alcohol (occasional), surgery, asthma	IgM, IgG	MS	Int.	-	Survived	6

^1^ Since the onset of symptoms, ^2^ quality was assessed in accordance with Murad et al. [54] through the appraisal of four domains (selection, ascertainment, causality of case, and reporting quality) that, in turn, were assessed by six dichotomous (i.e., 0/+1) items, for a total score potentially ranging from 0 to 6. See Appendix B, Table A2 for details. Note: AIHA: autoimmune hemolytic anemia; UAE: United Arab Emirates; DM: diabetes mellitus; HA: hemagglutinantion test; HPT: hypertension; TB: tuberculosis; EBV: Epstein–Barr virus; HBV: hepatitis B virus; ESRD: end-stage renal disease; CRD: chronic renal disease; OBS: obesity; ENC: encephalitis; OC: ovarian carcinoma; HEP; hepatitis; Class.: classification; CT: computed tomography; M: mild; MS: moderately severe; S: severe; Int.: interstitial pancreatitis; Nec.: necrotizing pancreatitis; PCR: polymerase chain reaction; NS1: non-structural protein 1; IgM: immunoglobulin M class; IgG: immunoglobulin G class; N.A.: not available; N.P.: laboratory work-up performed but details not provided.

## Data Availability

The original contributions presented in this study are included in the article. Further inquiries can be directed to the corresponding author.

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
