# Peer review of "Dengue and Acute Pancreatitis: A Systematic Review"

_tropicalmed, 2025, doi:10.3390/tropicalmed10120330_

Round 1

Reviewer 1 Report

Comments and Suggestions for Authors

Acute pancreatitis is a rare complication of dengue infection. In this systematic review, the authors have characterized the clinical spectrum dengue virus associated acute pancreatitis. Overall, the current review would shed much needed light on the etiology of dengue associated acute pancreatitis. However, the comments mentioned below needs to be addressed.

Major comment:

  1. Background section can be expanded to include the prevalence of acute pancreatitis specially in the regions included in the review (Asia and South America).
  2. It would be useful to include a forest plot to portray the multivariable analysis.
  3. What was the distribution of primary vs secondary dengue infection in the reports included in the study? Was the infection status (primary vs secondary) associated with AIP vs ANP since the secondary dengue infections tends to be more severe?

Minor comment:

  1. In figure 1, it is mentioned that 40 reports were excluded due to not fitting the criteria out of the 96. That should leave 56 reports. However, it says 66 in the final tab.
  2. Line 326: the sentence seems to be framed incorrectly and could be improved.

Author Response

Estimated Rev.1, 

first of all, thank you so much for your positive and constructive comments. We think that, by having addressed and replied to your suggestion, our paper has been ultimately improved in terms that could increase the likelihood to be eventually accepted in TropicalMed. More precisely, please accept our point-to -point comment:

1) Background section can be expanded to include the prevalence of acute pancreatitis specially in the regions included in the review (Asia and South America).

Reply: we agree with your suggestion, and the following section was therefore implemented in the introduction: "With an estimated annual incidence rate of 34.8 per 100,000 [31], acute pancreatitis has been considered a common disease with significant morbidity and mortality [32]. Conversely, DENV-associated pancreatitis has been considered a quite rare or even extremely rare condition [33,34]", while in the discussion we added the following statements: "

According to available estimates, acute pancreatitis is a quite common cause of hospitalization, and its incidence has increased during the last 50 years [32], currently ranging from 13 to 49 cases per 100,000 persons [31,32,52]. Global incidence peaks are associated with Eastern Europe (71.2 per 100,000/year; 95%CI 60.8 to 82.9), followed by North America (62.4 per 100,000/years, 95%CI 53.7 to 72.0), Andean areas of South America (45.8 per 100,000/year, 95%CI 39.6 to 52.9) [31]. Geographic areas most extensively represented in our pooled sample (i.e. Asia, and non-Andean areas from South America) are conversely associated with lower annual incidence rates (33.7 per 100,000, 95%CI 28.4 to 39.4; 38.1 per 100,000, 95%CI 31.6 to 45.9; 26.0 per 100,000, 95%CI 21.6 to 30.9; 20.1 per 100,000, 95%CI 17.7 to 22.7 for Central Asia, Southern Asia, Southeastern Asia, and Tropical Latin America) [31,32]."

2) It would be useful to include a forest plot to portray the multivariable analysis.

reply: we considered this option very interesting and consistent with our reporting strategy; the paper was therefore amended by including a specifically designed forest plot (Figure 4).

3) What was the distribution of primary vs secondary dengue infection in the reports included in the study? Was the infection status (primary vs secondary) associated with AIP vs ANP since the secondary dengue infections tends to be more severe?

Reply: this was a very interesting topic. We experienced substantial difficulties in addressing the proper dichotomization between cases of primary and secondary dengue cases due to the improper reporting strategy from most of parent studies; the following amendment was therefore included into the paper: 

"Fifth, only two studies included a formal definition of reported case as either primary or secondary dengue [67,118]. Even though the analysis of available serology suggests that a total of 14 cases could plausibly be considered secondary infections [58,61–63,66,67,77,80,83,87,98,110,116,118], this estimate likely represents an underestimation due to the inconsistent serology from several studies. As a consequence, the lack of association of secondary status with more severe features usually associated with ANP (20.8% vs. 19.6% of included AIP, p = 1.000) and the eventual death of the patient due to DENV-associated acute pancreatitis (all secondary cases had a generally favorable clinical outcome, p = 0.302; Appendix Table A3) should be quite cautiously appraised".

Classification in likely primary vs. secondary infection was also implemented within the main Tables of the study: please note that no substiantial differences were identified.

4) In figure 1, it is mentioned that 40 reports were excluded due to not fitting the criteria out of the 96. That should leave 56 reports. However, it says 66 in the final tab.

Reply: thank you, it was a mistake. The proper numbers are: 214 articles identidied, 108 screened, 96 seek for retrieval and eiligibility; 66 included in the final study.

5) Line 326: the sentence seems to be framed incorrectly and could be improved.

Reply: thank you, it was an annoying mistake. We double checked extensively the main text.

Reviewer 2 Report

Comments and Suggestions for Authors

This manuscript presents a systematic review of 70 published cases of acute pancreatitis associated with dengue virus infection, with particular attention to differentiating acute interstitial pancreatitis (AIP) from acute necrotizing pancreatitis (ANP). The review is registered in PROSPERO and conducted in accordance with PRISMA guidelines, using a comprehensive search strategy and a structured quality appraisal of the included case reports. The authors provide detailed clinical, laboratory, and imaging data, and explore predictors of ANP and mortality using regression and survival analyses.

The topic is clinically important and timely, given the global burden of dengue and the growing recognition of atypical and severe manifestations. Although the evidence base is limited to case reports and small case series—with the inherent methodological constraints that this entails—the authors acknowledge these limitations transparently.

Main comments:

The Discussion could benefit from a slight rebalancing. The limitations are clearly described—perhaps even overemphasized—while the strengths and added value of the review could be highlighted more explicitly, such as the fact that this represents the largest synthesis to date of DENV-associated pancreatitis with detailed clinical and imaging information.

A short, focused subsection on treatment and clinical management would be valuable from a clinical perspective. It would help to summarize how most patients were managed (e.g., conservative measures, invasive procedures, drainage of collections, supportive treatment), and to state clearly whether the dataset allowed the identification of any associations between management strategies and outcomes, or whether the heterogeneity and incompleteness of the data precluded such analysis.

In the Methods section, please clarify how missing data were handled in the statistical analyses (for example, listwise deletion, no imputation). In the Results and tables, ensure that denominators (N) are consistently reported for each comparison.

The high proportion of ANP in the sample is an interesting observation. It may be helpful to distinguish more explicitly between methodological explanations (such as publication bias or underdiagnosis of mild AIP) and potential biological mechanisms (including cytokine-mediated injury or microvascular ischemia), and to emphasize that the available data do not allow a definitive distinction between these hypotheses.

Given the use of different dengue diagnostic methods across the included reports (IgM ELISA, NS1, RT-PCR, hemagglutination), a brief and balanced discussion of their diagnostic performance in endemic settings would strengthen the manuscript. It may also be useful to recommend that future reports include molecular/NS1 testing and, where possible, serotype or genotype information.

Minor points:

A light language edit is recommended to correct minor typographical and stylistic issues (e.g., “workout” → “work-up”). Several long sentences in the Introduction and Discussion could be shortened to improve clarity.

The Conclusions could be made more clinically oriented by explicitly noting that acute pancreatitis should be suspected in dengue patients presenting with epigastric pain and significant enzyme elevations; contrast-enhanced CT is important for identifying ANP and collections; and mortality seems largely driven by the presence of collections and organ failure.

Subject to light improvements, this manuscript would make a valuable contribution to the literature on dengue-associated complications and offer practical guidance for clinicians in both endemic and non-endemic settings.

Author Response

Estimated Reviewer 2,

first of all, thank you so much for your comments: we are confident that, by identifying critical points from our study, and providing some appropriate and valuable suggestions for overcoming our shortcomings, our paper will be so properly implemented and improved to be acceptable for publication on TPID.

More precisely, please see our point-to-point reply: in the meanwhile, Sir, please accept our sincere thanks:

1) The Discussion could benefit from a slight rebalancing. The limitations are clearly described—perhaps even overemphasized—while the strengths and added value of the review could be highlighted more explicitly, such as the fact that this represents the largest synthesis to date of DENV-associated pancreatitis with detailed clinical and imaging information.

Reply: we thank you for this positive comment; we have revised some sections from discussion, and most notably the following lines: "Although the present study represents the largest synthesis to date of DENV-associated pancreatitis with detailed clinical and imaging information, several limits should be acknowledged, most of them".

2) A short, focused subsection on treatment and clinical management would be valuable from a clinical perspective. It would help to summarize how most patients were managed (e.g., conservative measures, invasive procedures, drainage of collections, supportive treatment), and to state clearly whether the dataset allowed the identification of any associations between management strategies and outcomes, or whether the heterogeneity and incompleteness of the data precluded such analysis.

reply: again, we agree - stressing that the present paper was not primarily defined for ascertaining medical treatment options. Therefore, we wrote: 

"

Even though our study was not specifically designed for providing detailed insights on the therapeutic options of DENV-associated acute pancreatitis, some details may be of certain interests for the daily practice of clinicians managing similar cases. First of all, only seven cases required any invasive approach for collection drainage [66,74,89,98,99,104,120], suggesting that without signs of multi-organ failure most of cases, including severe cases characterized by ANP and collections, could be conservatively managed, guaranteeing an appropriate alimentary support, and adequate volume replacement, being mostly affordable even in limited resource settings. Conversely, cases showing any need of respiratory support [35,65,71,73,80,89,92,103,113,119,120], or signs of renal failure [35,58–60,65,71,73,80,83,91,92,97,101,111,115] should be very cautiously monitored and could benefit from more aggressive therapies as more likely characterized by severe systemic complications and a dismal prognosis."

3) In the Methods section, please clarify how missing data were handled in the statistical analyses (for example, listwise deletion, no imputation). In the Results and tables, ensure that denominators (N) are consistently reported for each comparison.

Reply: we again agree and performed a further double checking of source studies. More precisely, the text was amended as follows:

a) "studies were deemed eligible for inclusion in the systematic review even when the specific diagnostic test was not explicitly reported, provided that the original source confirmed a laboratory-established diagnosis (i.e., ‘not provided’, NP)"

b) "Missing values for assessed variables were handled by means of listwise deletion, i.e. observations with incomplete data were excluded from the specific analysis in which the missingness occurred".

4) The high proportion of ANP in the sample is an interesting observation. It may be helpful to distinguish more explicitly between methodological explanations (such as publication bias or underdiagnosis of mild AIP) and potential biological mechanisms (including cytokine-mediated injury or microvascular ischemia), and to emphasize that the available data do not allow a definitive distinction between these hypotheses.

Reply: we totally agree with this comment, and the comment section was accordingly revised:

"The oversampling of ANP cases compared to AIP could be explained in terms of potential publication bias or by means of biological mechanisms, considering that available data do not allow any definitive distinction between these two hypotheses. On the one hand, we cannot rule out the oversampling of more severe and complicated cases due to the design of our research strategy, based on case reports, with a larger proportion of AIP cases simply undiagnosed as characterized by mild signs and symptoms, or even misdiagnosed"

5) Given the use of different dengue diagnostic methods across the included reports (IgM ELISA, NS1, RT-PCR, hemagglutination), a brief and balanced discussion of their diagnostic performance in endemic settings would strengthen the manuscript. It may also be useful to recommend that future reports include molecular/NS1 testing and, where possible, serotype or genotype information.

Reply: we agree with your suggestion, and some information on the diagnostic performance was added as follows:

"

For instance, according to a recent systematic review with meta-analysis [152], IgM ELISA is characterized by poor diagnostic accuracy (Sensitivity 62%, 95%CI 45 to 75; Specificity 85%, 95%CI 76 to 91), particularly during the early stages of the syndrome (Sensitivity 17%, 95%CI 3 to 51; Specificity 84%, 95%CI 47 to 97 for days 0 to 4). On the contrary, NS1 testing usually shows high diagnostic accuracy, comparable to RT-qPCR, in both early stages of the symptomatic infection (pooled sensitivity for days 0 to 4 since the onset of symptoms: 90%, 95%CI 68 to 98 and 95%, 95%CI 77 to 99 for NS1 testing and RT-qPCR, respectively; pooled specificity of 93%, 95%CI 71 to 99, and 89%, 95%CI 60 to 98, for NS1 testing and RT-qPCR, respectively), and in the whole of symptomatic subjects (pooled sensitivity of 85%, 95%CI 76 to 91 and 86%, 95%CI 63 to 93 for NS1 testing and RT-qPCR, respectively; pooled specificity of 90%, 95%CI 83 to 94, and 96%, 95%CI 84 to 98, for NS1 testing and RT-qPCR, respectively). As more than half of the included cases were diagnosed on IgM ELISA, the actual diagnosis of the included cases should be carefully addressed. Subject to the financial resources of the healthcare systems involved, a more rigorous and systematic use of diagnostic panels incorporating both serological assays and molecular/NS1 testing is therefore desirable".

6) A light language edit is recommended to correct minor typographical and stylistic issues (e.g., “workout” → “work-up”). Several long sentences in the Introduction and Discussion could be shortened to improve clarity.

REPLY: we agree with your suggestion, and several sentences were accordingly revised; for example, section 1.2 was revised as follows: 

Dengue is currently acknowledged as the most common mosquito-borne tropical viral disease in humans [1,2,5], as well as one of the most reported travel-associated infectious diseases [15]. Although substantial uncertainty due to the high proportion of asymptomatic infections and the inadequate reporting systems from several geographic areas [1,5,16,17], around 50% of all human beings are nowadays considered at risk for developing dengue infection due to the four main DENV serotypes (DENV 1-4) [10,12,13]. 

7) The Conclusions could be made more clinically oriented by explicitly noting that acute pancreatitis should be suspected in dengue patients presenting with epigastric pain and significant enzyme elevations; contrast-enhanced CT is important for identifying ANP and collections; and mortality seems largely driven by the presence of collections and organ failure.

REPLY: we agree with your suggestion, and conclusions were therefore revised as follows:

Acute pancreatitis associated with DENV infection has been increasingly reported, suggesting that pancreatitis may represent a true complication of dengue rather than a serendipitous association. It should be suspected in dengue patients presenting with epigastric pain and significant enzyme elevations; contrast-enhanced CT is important for identifying ANP and collections. Even though the available evidence is limited, acute pancreatitis associated with dengue infection is characterized by relatively high case-fatality rate, and higher occurrence of ANP compared to AIP. DENV-associated acute pancreatitis should be acknowledged as a potentially severe condition, with mortality driven by collections and organ failure. Nonetheless, substantial overestimation of its actual severity cannot be ruled out. Due to the increasing global burden of DENV infection and the significance from both clinical and public health point of view of acute pancreatitis as potential complication, our data stress the importance of a more appropriate definition of the actual epidemiology of this condition by means of studies specifically targeting the occurrence of acute pancreatitis during the clinical course of DENV infection.

In the end, we would thank you again for the very positive appraisal of our paper and the constructive suggestions you shared with us.